Multi-objective federated learning traffic prediction in vehicular network for intelligent transportation system

http://orcid.org/0000-0001-9602-9059 Aalavanthar Arulmurgan 1
S. Famila 2
http://orcid.org/0000-0001-6935-4980 Sundaramurthy Shanmugam 1 shanmugam.network13@gmail.com
http://orcid.org/0000-0003-0201-2753 Cirillo Stefano 3 scirillo@unisa.it
http://orcid.org/0009-0000-6627-8820 Solimando Giandomenico 3
Polese Giuseppe 3
1 Department of Computing Technologies, SRM Institute of Science and Technology, Kattankulathur , Chennai, Tamilnadu , India
2 Department of Electronics & Communication Engineering, Vel Tech Rangarajan Dr. Sagunthala R&D Institute of Science and Technology , Chennai, Tamilnadu , India
3 Department of Computer Science, University of Salerno , Fisciano, Salerno , Italy
Wang Wei
Electronic publication date: 2025 Jun 3
Publication date: 2025
Volume: 11
Electronic Location ID: e2922
Received 2024 Sep 25; Accepted 2025 May 6
Copyright: © 2025 Aalavanthar et al.
Copyright year: 2025
Copyright holder: Aalavanthar et al.
License: This is an open access article distributed under the terms of the Creative Commons Attribution License, which permits unrestricted use, distribution, reproduction and adaptation in any medium and for any purpose provided that it is properly attributed. For attribution, the original author(s), title, publication source (PeerJ Computer Science) and either DOI or URL of the article must be cited.
License URL: https://creativecommons.org/licenses/by/4.0/

Keywords: Federated learning, Traffic prediction, Vehicular networks, Intelligent transportation systems (ITS), Multi-objective optimization, Machine learning

Funding: The authors received no funding for this work.

==============================
The spatial-temporal data of future freight traffic speed in the metropolitan region must be properly understood to develop freight-related traffic management strategies. This work introduces a new approach to traffic prediction using multi-objective federated learning. Instead of relying on a centralized cloud server for data processing, collaborative training is implemented among several participants. The proposed method utilizes the advantages of reinforcement learning in dynamic decision-making scenarios and the expressive capabilities of graphical models to identify traffic intensity. Furthermore, a new methodology integrates federated learning concepts with multi-objective optimization to forecast traffic patterns accurately. The proposed approach exhibits a higher level of performance than existing methods for estimating traffic speed. It achieves a communication delay of 23.4%, packet delivery ratio (PDR) of 92.45%, packet loss rate of 12.34%, prediction accuracy of 97.45%, and resource utilization of 89.56%. The visualisation findings demonstrate that this new approach is able to successfully capture interconnections of metropolitan areas in different neighboring cities.

Introduction

Telecommunications networks beyond 5G must satisfy stringent requirements for latency, security, and reliability. As the number of devices in these networks grows, the volume of data transmitted increases, necessitating the development of advanced techniques for network monitoring and management (Sarker, 2022). The evolution of 5G requires innovative methods for adapting network protocols and managing resources across diverse services. Artificial Intelligence (AI) encompasses systems and techniques capable of perceiving their environment and taking actions to optimize the achievement of specific objectives. This approach is increasingly viable for the design of sophisticated communication networks (Salameh & El Tarhuni, 2022). AI incorporates all components contributing to the enhancement of machine intelligence, while machine learning (ML), a subset of AI, refers to systems capable of autonomously acquiring knowledge and refining their performance through algorithmic processes. ML enables the development of intelligent systems that can learn and adapt without human intervention, and it is employed extensively in contemporary AI applications due to its ability to process large volumes of data (Abdellah & Koucheryavy, 2020; Breve, Cimino & Deufemia, 2024). By analyzing new data, ML generates insights, predicts outcomes, and addresses complex challenges such as wireless network optimization and attack detection.

The Internet of Things (IoT) facilitates infrastructure, application, and security innovations by leveraging intelligent edge data to enhance operations and user experiences (Singh, 2021; Anisetti et al., 2022; Cimino & Deufemia, 2025). IoT systems use sensors to connect everyday devices, enabling them to collect and transmit data. As the number of IoT devices grows, so does the volume of data generated, leading to the development of specialized services for operators across various industries. IoT devices continuously establish connections and collect vast amounts of data (Abdellah et al., 2021; Ardagna et al., 2023). These systems are increasingly capable of triggering actions based on predefined conditions or in response to data from diverse applications. While IoT devices are adept at gathering and transmitting data, human oversight remains essential for analysing this data, extracting valuable insights, and developing intelligent applications. To achieve advanced automation, smart IoT devices must possess the functionality to allocate resources, enable communication, and deliver network services (Rekkas et al., 2021).

AI can play a pivotal role in designing the physical layer, making complex decisions, managing networks, and optimizing resources in 5G mobile systems (Nassef et al., 2022). Moreover, big data analytics can enhance the understanding of 5G cellular networks and their operational characteristics (Abubakar et al., 2020). ML is widely applied across various sectors due to its capacity to process large datasets and identify intricate patterns that would be difficult to detect manually. Through statistical analysis of extensive data, intelligent systems can provide optimal solutions for a variety of challenges, including fault detection in systems such as heat supply networks (Tsourdinis et al., 2022) and rail infrastructure monitoring, thus reducing the risk of accidents. Furthermore, ML is particularly effective in time-sensitive domains such as healthcare, security, and emergency response, where timely identification and reporting of critical events are vital for informed decision-making (Abdellah & Koucheryavy, 2022; Cirillo, Solimando & Virgili, 2024). ML is also widely used in antivirus software, petroleum reservoir research, and business analytics (Kaur et al., 2021).

Reinforcement learning (RL) enables traffic prediction systems to develop optimal strategies through interaction with the environment. Unlike static models or rule-based systems, RL updates its decision-making process based on real-time feedback to adapt to dynamic traffic conditions. RL agents can forecast traffic congestion, adjust signal timings, and recommend alternate routes by optimizing long-term objectives, such as reducing travel time or fuel consumption. Its ability to handle complex, high-dimensional environments makes RL particularly suitable for dynamic transportation systems. Multi-objective federated learning (MOFL) addresses the challenges of traffic prediction by preserving privacy and enhancing scalability. Traditional centralized traffic prediction models rely on aggregating large datasets from multiple sources, which can raise concerns about privacy and efficiency (Caruccio et al., 2024). In contrast, MOFL decentralizes data processing, keeping sensitive location-based data on local devices while optimizing a global model. MOFL improves the accuracy and fairness of predictions by integrating localized information, optimizing multiple objectives like reducing congestion, minimizing travel time, and mitigating environmental impact. This decentralized approach also reduces communication costs, ensures real-time updates, and decreases energy consumption, making it an ideal solution for modern traffic prediction systems. Despite advances in traffic prediction methodologies, existing models remain constrained by privacy concerns, scalability challenges, and the inability to adapt in real time to fluctuating traffic patterns. Traditional centralized approaches require data transmission to a central server, which poses security risks and computational inefficiencies. In addition, static models fail to capture the dynamic nature of urban mobility. This article addresses these issues by introducing MOFL, which integrates reinforcement learning, federated learning, and graphical neural networks to enhance prediction accuracy while preserving data privacy.

This suggests the following main contributions: Graphical reinforcement learning (GRL) can be an effective approach for traffic intensity detection, leveraging the strengths of reinforcement learning in dynamic decision-making environments and the representation power of graphical models.

GNNs can be integrated into the reinforcement learning framework to handle graph-structured data. GNNs can process the graph, capture the spatial dependencies between nodes, and provide meaningful embedding for the RL agent.

Federated learning (FL) enables decentralized devices to train a model while retaining data locally. Data privacy is especially important in situations like traffic data from different cities.

The remainder of the article is organized as follows: “Related Work”, we describe relevant studies concerning the applications of Federated Learning, in the context of traffic prediction in vehicular networks. “Preliminaries” provides a brief overview of the preliminaries of the task model. “Proposed Methodology”, presents the proposed method, including the description of the transmission of information among the entities and the integration with the federated learning. “Experimental Evaluation” shows the results obtained by the experimental evaluations. “Limitation of Proposed Method” discusses the limitations of the proposed approach, including the challenges faced during implementation. Finally, conclusions and future directions are provided in “Conclusion and Future Works”.

Related work

This section specifically addresses the efficient method to forecast traffic patterns in vehicle communications, which may be used in real-time applications, such as dynamic traffic management. RSUs can capture data on the speed and flow of traffic. This information may then be sent to a control unit, which uses it to automatically calculate the amount of traffic.

Federated learning-based traffic prediction

According to Gautam et al. (2022), combining feedforward neural networks with the Quasi-Newton approach for optimization improves traffic flow forecast accuracy. Utilizing the Lagrange multiplier and Jacobian vector minimizes the error factor. A decentralized federated learning-based spatial-temporal model for freight traffic speed prediction is suggested in Turki (2024). Instead of processing data on a cloud server, participants collaborate on training. The mean improvement for MAE, RMSE, and MAPE is 8.3%, 8.2%, and 8.6%, respectively. Shen et al. (2024) introduces a novel decentralized federated learning approach with a spatial-temporal transformer network that improves freight traffic speed forecasting without requiring a centralized server. The model enables collaborative training among multiple participants while preserving local personalization. Testing on real-world data from the Nanjing Metropolitan Area showed significant improvements over existing methods 8.3% MAE, 8.2% RMSE, 8.6% MAPE, while effectively capturing spatial-temporal dependencies across urban regions to inform proactive traffic management strategies. In Yaqub et al. (2024), a novel intelligent traffic flow forecasting model called MFVSTGNN is provided. Space-temporal graph-based deep learning is used in the Multilevel Federated Learning model. Variational graph autoencoder (VGAE) dynamically generates a spatial and semantic adjacency matrix in the first step. This matrix saves important data for prediction accuracy. The next step involves predicting using a broad spatial-temporal graph neural network. FedAGCN, a federated learning system using asynchronous graph convolutional networks, is introduced in Liu et al. (2023). It seeks to accurately anticipate traffic flow in real-time. The study (Qi et al., 2023) introduces a two-layer federated learning strategy for 6G’s distributed end-edge-cloud architecture. While protecting data privacy and reducing communication overhead, our methodology improves learning speed and accuracy. The federated learning technique optimizes local and global contexts of cars and RSUs in 6G vehicular networks using a novel multi-layer heterogeneous model selection and aggregation mechanism. The SDN-NFV system model is shown in Zhou et al. (2021). This paradigm includes NFV-enabled edge FL (eFL) aggregation servers and device/resource abstractions. This improves automation and control. MADQNs deploy a self-learning software system, improve resource allocation rules, and encourage computation offloading. The innovative VeNet hybrid learning system was created in Tam et al. (2022). It uses a stacked AE with a central and several local AEs. A unique hybrid VeNet learning system training approach is presented. The VeNet hybrid learning system uses geographic and temporal correlations in mobile vehicle data to accomplish spatial learning (Vargas-Solar et al., 2024).

Traffic prediction

In Wang et al. (2023) provided a hybrid methodology that integrates historical data with road networks. Timely and accurate traffic flow forecasting is crucial for determining successful traffic control tactics and directly influences the efficacy of traffic management measures. In Guo et al. (2023) authors provided a scenario engineering-based calibration and validation methodology for autonomous vehicle trajectory prediction to better evaluate the system in demanding circumstances. The vectorisation method extracts contextual elements from the scene and agent trajectory information from the HD map, and the graph neural network models high-order interactions to predict interactive trajectories. In Sepasgozar & Pierre (2022), the authors propose a technique to anticipate network traffic using road traffic characteristics. This model uses RF-GRU-NTP to estimate network traffic flow based on road and network traffic concurrently.

Reinforcement learning-based traffic prediction

Instead, in Boukerche, Zhong & Sun (2021) authors propose an RL-based cooperative traffic signal management strategy for traffic road networks with data transmission delays. This proposed method was tested in synthetic and real-world circumstances with varied data transmission delays. Using deep reinforcement learning (DRL), an adaptive traffic light management system is constructed in Mushtaq et al. (2021) to maximize traffic flow at junctions during peak traffic times. In the second phase, we use smart re-routing (SR) for junction traffic. To avoid congestion at road crossings, re-route heavy vehicles onto different pathways. In Li et al. (2023), a deep reinforcement learning-based controller optimizes velocity under traffic lights and uncertainty limitations to reduce energy consumption and ensure driving safety. The aforementioned poll highlights some shortcomings as follows: Decentralized federated learning necessitates frequent communication among vehicles for model updates. This can result in high bandwidth usage, which may be problematic in scenarios with limited network resources.

Asynchronous updates in Federated Learning and Asynchronous Graph CNN (FLAGCN) require frequent exchanges of node embeddings and model parameters between vehicles and possibly central servers. This can significantly increase the communication load, particularly in dense traffic scenarios.

Designing and training deep Q-networks for multiple agents involves complex model architectures and training procedures. Ensuring robust convergence and effective learning in MADQNs requires careful tuning of hyperparameters and optimization strategies.

To overcome the aforementioned challenges, the multi-objective federated learning traffic prediction is designed in this work, which offers significant advantages, including privacy preservation, data diversity, decentralized learning, reduced communication overhead, collaborative model improvement, resilience to network dynamics, and regulatory compliance (Khan et al., 2024). These advantages make it a promising approach for enhancing traffic management, optimizing resource allocation, and improving user experience in smart vehicular environments. Graphical models can capture complicated traffic system interactions in an organized and probabilistic manner, making them important for traffic intensity detection. Due to flow continuity or congestion propagation, traffic intensity typically relies on nearby areas. Graphical models clearly reflect spatial interdependence, improving pattern recognition.

Preliminaries

It is presumed that the road segment is composed of k links, with each link representing a distinct portion of the road. Every section should have a single RSU. The historical database has information for a certain number of days, represented by i=1,2,…,nj, and j=0,5,10…,nj. This information is recorded in 5-min intervals. The variable t represents the predicted time on the prediction day p. Then, ts is the traffic pattern’s start time in the previous days. The velocity at link k on prediction day p at time t−j is v(i,t−j,p). Similarity is expressed by h, which might be 1, 2, and so on. The number of days in the history database is nh. On historical days, ts reflects the traffic pattern start time. Traffic volume at connection k at time ts−j on the historical day h is v(i,ts−j,h).

The primary factor influencing travel time on a route is the level of congestion experienced on that road. This congestion is often caused by bottlenecks, which can occur regardless of time or place. To account for these bottlenecks in our analysis, we assign weights to different road segments based on their velocity profiles. Higher weights are allocated to sections with lower speeds, as these slower segments constitute bottlenecks in the network. Building upon these weight calculations, the method of pattern matching involves finding historical patterns or patterns that are most comparable to one another. These patterns are evaluated based on the weighted features we’ve established, allowing for more accurate similarity assessments. Equation (1) uses the following practical formula:

(1) w(i,j)=1[v(i,t−j,p)]C

where C is the constant, and the search is projected to take minutes on historical days, ensuring that t+x≥ts≥t−x.

Therefore, the main objective is to create a database of past days. An approach to looking for these patterns involves examining the whole history record to identify the most comparable pattern. However, this method requires significant processing resources and is time-consuming. Thus, traffic pattern similarities are the squared difference between the projected temporal and historical traffic patterns. History’s traffic pattern is the most comparable, with the lowest sum of squared differences. Equation (2) is used to obtain the formula for the goal function in constructing the traffic patterns.

(2) Δ2(p,t,h,ts)=∑i=0nt∑j=0njw(i,j).L(i)L[1v(t,t−j,p)−1v(t,tj,h)]2

where the traffic weight in the cell (i,j) is w(i,j), section length is L(i), and road stretch length is L. The squared difference between present and historical patterns is Δ2(p,t,h,ts).

Although several studies have explored federated learning for traffic prediction, most fail to balance communication efficiency with real-time adaptability. Recent advances in graph neural networks (GNNs) and reinforcement learning (RL) show promise in capturing spatial and temporal dependencies in vehicular networks. The proposed MOFL model builds on these approaches, ensuring decentralized learning while optimizing multiple traffic-related objectives simultaneously.

Proposed methodology

The system architecture delineates the comprehensive framework of the vehicular ad-hoc network (VANET) system, including the transmission of information among the entities within the VANET system, as well as the integration of federated learning with the VANET architecture. The following part provides a concise explanation of the system model and federated learning in VANET, as seen in Fig. 1. FL protects traffic data from different locations by decentralizing and securing critical information during training. Local traffic data, including car counts, congestion levels, and GPS traces, never leaves FL. Instead, models are trained on local data, and only encrypted or anonymized model updates (e.g., gradient or weight modifications) are aggregated by a central server. Secure aggregation combines updates without disclosing individual contributions, whereas differential privacy adds noise to updates, making crucial facts computationally hard to discern.

Figure 1 Proposed system model for VANET.

Network model

The VANET system concept primarily consists of three key entities: roadside unit (RSU), vehicle registration, and RSU registration.

Roadside unit

An RSU refers to a computer device that is located along the roadside or at specific locations, such as parking lots or crossroads/junctions. The goal of this device is to establish a local connection with automobiles that are within its range. The network devices included in RSUs use IEEE 802.11p technology for their localized and specialized communication across short distances. Every RSU is interconnected with adjacent RSUs via a wired network and with the vehicle user through a wireless network.

Vehicle node

Vehicles have an On-Board Unit with advanced communication devices, wireless transmission modules, and a trusted execution environment. Vehicles can establish communication with other vehicles and RSUs and are equipped with sufficient computational capacity to do basic computations. The computations detect and report road condition events and determine the target’s direct and indirect reliability using the data.

Activities in the network

When a vehicle node detects traffic congestion, crashes, bad weather, and so on, it sends event data to the RSU. The latter sends event data to the surrounding vehicle nodes to confirm it. An occurrence in a distant area without RSU coverage will be relayed directly to nearby vehicles for joint verification. Upon receiving the verification request to observe the target event, the vehicle node can provide several observation reports that vary in terms of speed, distance, and other characteristics. Let us assume that the network has already defined a set of basic event structures, denoted as β={e1,e2,…,en}, where each e1 represents an event that can be detected by vehicles. Herein, Wi, the vehicle nodes’ reports, may be described as a collection of fundamental events, Ei. These reports additionally provide the vehicle’s status when it witnessed an incident, such as timestamp and distance di. The whole report format is Wi={Ei,ti,di,…}.

Graphical reinforcement learning for traffic intensity detection

We propose a GRL method to detect traffic intensity in vehicle networks. Our system represents the road network as a graph and uses reinforcement learning to make smart decisions about traffic monitoring. The approach uses two main parts: an Actor network that decides what actions to take, and a Critic network that evaluates how good those actions are. This helps vehicles automatically monitor traffic while using communication resources efficiently. The system works in two simple steps: first, collecting traffic data during normal operation, then using that data to improve the model’s performance. The GRL for environment scanning in VANETs is an innovative approach that can significantly enhance the efficiency and effectiveness of data collection and processing in these networks. It involves collecting and analysing data from the surrounding environment to improve decision-making processes. This includes detecting obstacles, monitoring traffic conditions, and gathering information about road and weather conditions. GRL is a subset of RL that incorporates graph structures to represent the state and action spaces. This approach is particularly useful in scenarios where relationships between entities can be naturally represented as graphs, such as in VANETs. This graph structure serves as the foundation for our reinforcement learning system to make better decisions within the vehicle network. The network that the controller is linked to is represented by the Environment. The controller gets the network state from the environment and generates a policy using the Actor neural network’s OnlinePolicy. The previous state, the controller action, the environment reward value, and the recently observed state constitute a four-tuple record Laststate,action,reward,netstate. The data is preserved for neural network model training in the experience reservoir. Our approach uses two separate neural networks to handle different aspects of the decision-making process in this graph environment. It uses two networks. The Critics utilize the online Q technique. An actor NN using the OnlinePolicy’s decision network. The OnlinePolicy follows the state-action paradigm to make decisions μ depending on the present environment. However, OnlineQ estimates the reward of the environment for the controller’s choice. The state,action combination reward.

The OnlinePolicy and the OnlineQ have similar target networks, target policy and targetQ. Soft updates aid in convergence and avoid gradient fluctuations during training. Actor and Critic NNs often have an online and target network. The actor neural network receives real-time actions via OnlinePolicy. Target policy updates the Critic neural network.

Both OnlineQ and targetQ produce the same output, which is the value of a single state. However, their inputs are different. OnlineQ requires the input of the specific actions performed by the NN actor and the current state. While the Actor network decides what to do next, the Critic network judges how good those decisions are, creating feedback that helps improve future decisions.

TargetQ utilizes the results generated by the Target policy. Figure 2 displays the structural diagram of graphical reinforcement learning using Actor NN and Critic NN. It shows the Actor NN at the top and the Critic NN at the bottom. Moreover, it shows tensor transformation processes in blue rectangles, neurons in golden circles, and tensors in grey circles.

Figure 2 Graphical reinforcement learning for vehicular environment.

The Actor model takes four inputs: the network’s topological adjacency matrix A, the current features X of all nodes, the one-hot code l for the data packet’s current location, and the destination node d.

When transferring data packets, it is necessary to take into account both the status of the present node and the state of its neighbouring nodes. The Actor uses a graph CNN to extract and combine the state characteristics of the present node and its neighbouring nodes. After collecting the network state, it is essential to examine the destination node due to its strong correlation with data forwarding. Thus, we compress the output of graphical layers and combine it with l, d. The vector state represents network environment state properties, as supplied by:

(3) state=[flatten(GCN(A,GCN(A,X))]

where the graph convolution operation is GCN. The Hadamard product operation is used to calculate the output μ of the Actor model, combining fully connected layer results with an inner product operation. Softmax activation function is utilized at this layer, with output representing the next hop policy μ for the current data packet. The Critic has five inputs. The Actor model encompasses four input parameters, as well as the next hop action, denoted as “ a”. GCN layers are included in the Critic model to extract characteristics from the controller and its neighbouring nodes. The GCN layer weights, however, are not trained. Direct replication of the Actor model’s weights from the corresponding GCN layers occurs. The goal is to ensure that all nodes’ state attributes that have been collected by both models are completely consistent. The layer of the Critic model is entirely connected. The topmost layer employs a solitary neuron to accommodate the feedback value Q of the action policy implemented by the model. The traffic control model, which is built upon deep graph reinforcement learning, consists of two distinct stages. There are two stages in the process: the data collecting stage, which occurs while the simulation network is operating, and the training stage. Every node in the network follows the controller’s instructions and chooses the next hop in accordance with the WSN’s current status. Following the completion of the forwarding process, the next node will record the current state and reward value in the experience pool for training, combining them into a four-tuple record Laststate,action,reward,netstate. Algorithm 1 gives a thorough description of the process.

Algorithm 1 Algorithm of the proposed process.

 1: Input: batch size, θμ∗	
 2: Load the weights: θμ∗ for target model	
 3: repeat	
 4:    Generate data packet p, calculate its next hop a using θμ∗ and stochastic process, then record it with the current network status: p:={destination,location,state,a,data}	
 5:    while length(experience replay) <batchsize do	
 6:       if packet p received then then	
 7:          if p has arrived at its destination then	
 8:            Done ← true	
 9:       else	
10:            Done ← false	
11:       end if	
12:       Add experience replay list with {p→state,p→a,r,newstate,done}	
13:       generate new state and compute reward r for p→a	
14:       p→state:=newstate;p→location:=location	
15:     end if	
16:     if not done then then	
17:        Use θμ∗ to find the next hop a	
18:         p→a:=a	
19:     end if	
20:     Transmit data packet p	
21:   end while	
22: until True	

This combined approach solves the challenge of detecting traffic conditions in real-time by building on recent advances in graph networks and reinforcement learning. Although several studies have explored federated learning for traffic prediction, most fail to balance communication efficiency with real-time adaptability. Recent advances in GNNs and RL show promise in capturing spatial and temporal dependencies in vehicular networks. The proposed MOFL model builds on these approaches, ensuring decentralized learning while optimizing multiple traffic-related objectives simultaneously.

Secured federated learning system

In this research, we use a machine learning framework to enhance the accuracy of network intrusion categorisation by including data from many nodes. The core architecture of our system comprises a single server and k clients (C1,C2,…,CK) that have restricted computational capacity.

Both the clients and the server use a common neural network architecture and learning goals. Additionally, each client maintains its local model, which may be trained using its input. The server picks a subset of clients and provides the newest global model parameters at the start of each training cycle. Wireless communication helps clients and servers share data. We assume that customer data types and quantities fluctuate in the simplified system. The computational capability of each client varies, as does their connection latency with the server. Additionally, clients may become non-functional throughout the training process. Furthermore, it is acknowledged that the global model is updated at the server end, while the local model is changed at the client end. At the start of the training phase, either all or a subset of the clients are selected, and the most up-to-date global model parameters are supplied to them. Using the collected local data DK,CK does optimization over several iterations using techniques, such as adaptive moment estimate (Adam). The revised local model parameters are of

(4) at+1=at−δ(at)

where δ(at) represents the batch gradient, and δ denotes the learning rate. Subsequently, the changes are communicated to the server, where a secure aggregation process is performed.

(5) at+1←wt−∑k=1Knknat+1k.

In this FL model, each of the K clients trains a local model using a common global model. However, instead of training on a central server, the local models are trained on different local datasets. Afterwards, they transmit the knowledge acquired from these regional training sessions to a central server over a secure SSL/TLS-authorized connection managed by the communication administrator. The aggregation server updates the global model with optimal parameters. The starting weights are “ w” and the number of FL rounds, until convergence is reached, is “R”. During communication round t, the server receives the weight of each client, which is supplied according to Eq. (4) derived from the FedAvg algorithm.

Equation (5) utilizes n for the total number of client datasets and nk for each dataset. Iteration changes the global model to at+1. Clients with active, charging devices and unmetered Wi-Fi networks are selected by the server before FL begins. Then, the system’s many elements complete the task: At time t=0, the server creates an NN model based on a global intrusion detection model. Currently, the count is made for the number of hidden layers, neurons, epochs, and other relevant factors. The symbol w represents the starting weight of the model.

Each client, denoted as k(k∈[1,2,...,K]), is required to download and use the global model, independent of their participation in the FL process. Using their private data, each of the K clients independently trains the global model in parallel and generates a new set of local weights denoted as at+1k.

Designated clients utilize device data to improve the model while maintaining the anonymity of local data.

Only improved model parameters for increased intrusion detection are communicated to the central server to protect client privacy.

After receiving all the modifications, the server merges the weights from different node models to provide an enhanced model (Eq. (6)). The FedAvg technique is used for the aggregation process. This approach evaluates the parameters depending on the dataset’s size at each node.

The central server sends the revised model parameters to the clients.

Each customer utilizes the updated model parameters and modifies them according to the new data.

Steps 4, 5, 6, and 7 are repeated to continue the process of model learning and development.

The FL parameters provide intricacy to FL compared to a traditional ML system due to the unique environment they reflect. The diverse data skewness across client devices is crucial. In a multi-class classification problem, a significantly imbalanced data situation might occur, where each client is associated with just one data label. In MOFL, communication load is the volume of data required for transmission between local clients (cities) and the central server during the model training process. When working with large-scale traffic prediction systems dispersed over many cities, it is essential to optimize the communication burden in MOFL. Because hyperparameters have a major impact on the model’s performance, convergence rate, and communication effectiveness, choosing the right ones is essential. The proper hyperparameters may aid in striking a balance between controlling communication costs and achieving numerous goals (including reducing traffic, travel time, and fuel consumption), particularly in dispersed contexts like traffic prediction across many cities.

Experimental evaluation

This section contains the experimental methodology for our proposed method, along with a comprehensive analysis of its performance. The performance criteria used to examine a suggested traffic forecast technique are essential for evaluating its accuracy, efficiency, and robustness. These scores generally indicate the method’s capacity to accurately forecast traffic conditions across diverse situations. The choice of convergence criteria and iteration rounds in an FL framework, particularly for MOFL in traffic prediction, is essential for optimizing model performance, communication efficiency, and computing overhead.

Dataset overview

Urban traffic is increasingly becoming a global challenge. Factors such as rapid urbanisation, ageing infrastructure, inefficient traffic signal management, and limited access to real-time data exacerbate the issue. The consequences are substantial: in 2017 alone, traffic congestion cost U.S. commuters 305 billion, including expenses from wasted fuel, lost productivity, and increased freight costs, as reported by INRIX. Given the physical and economic constraints of expanding road networks, cities must adopt innovative technologies to enhance traffic management and effectively alleviate congestion.

For this study, we utilized an authentic VANET dataset based on the 802.11 ad-hoc network type. This dataset incorporates two communication modes, Vehicle-to-Vehicle (V2V) and Vehicle-to-Roadside (V2R), employed to evaluate short-range communication on highways. The data was collected using external antennas mounted on the rooftops of vehicles. GPS data provided parameters such as longitude, latitude, speed, and direction at 2-s intervals. The dataset was gathered on a six-lane highway in Atlanta, comprizing five regular lanes and one high-occupancy vehicle (HOV) lane. Data collection occurred between 2 PM and 5 PM. The GPS positional accuracy was estimated to be within five to seven meters, achieved through interpolation.

V2V communication was evaluated using designated vehicles, while V2R tests were conducted with mobile vehicles and a Road-Side Unit (RSU) positioned on an elevated bridge. The V2R connection involved transmitting 1,470-byte packets at approximately 150 packets per second.

The CAR_HACKING dataset (https://ocslab.hksecurity.net/Datasets/car-hacking-dataset) a includes various types of attacks, such as DoS assaults, fuzzy attacks, driving gear spoofing, and RPM gauge spoofing. A detailed description of the dataset is provided below. The Dos attack involves the transmission of CAN ID messages ‘0000’, which are the most prevalent messages, sent every 0.3 milliseconds.

The Fuzzy attack is characterized by the injection of messages with entirely random CAN ID and DATA values every 0.5 milliseconds. The Spoofing attack (RPM/gear) includes the transmission of precise CAN ID messages related to gear and RPM data at intervals of one millisecond. The dataset includes the following data properties: the Timestamp, which records the time in seconds; the CAN ID, which is the identifier of the CAN message in hexadecimal format (e.g., 043f); the DLC, which represents the number of data bytes and ranges from 0 to 8; the DATA values, represented as DATA{0} to DATA{7}, each indicating a byte; and the Flag, which identifies whether the message is injected (T) or normal (R).

Experimental setup

The experimental evaluations have been conducted using the datasets shown in “Experimental Evaluation”. The models were created on a workstation with an Intel i9 CPU at 5 GHz, 14-core, and 64 GB of memory, equipped with a GPU NVIDIA 3060 GPU, and by using PyTorch 1.9.0, Python 3.8.11, and PyCharm 1.3. The experimental session was conducted on Google Colaboratory using Python 3 as the programming language. This technique utilizes NumPy and other established libraries, using multi-dimensional arrays and matrices for implementation. We made use of Pandas’ robust facilities to manipulate and analyse data structures. Furthermore, TensorFlow and Keras are used for ML and DL tasks. Moreover, Scikit offers a wide range of supervised and unsupervised machine-learning algorithms.

Experimental results

The graph below shows the forecast results for the following hour, day, month, and year of the proposed ways to count automobiles. Figure 3 illustrates a timing diagram of many vehicles in hours, days, years, and dates using the proposed approach (Fig. 4).

Figure 3 The timing diagram of several vehicles relative to the time in hours, days, and years, including dates for the planned approach at the junction.

(A) Shows the count of traffic by junction (2015–2017); (B) shows the weekly traffic patterns by junction; (C) shows the hourly traffic volume by junction; (D) shows the monthly traffic fluctuations by junction; (E) shows the seasonal traffic trends by junction; (F) shows the yearly traffic growth by junction.

Figure 4 Vehicle count at each junction by hour, day, month, and year.

(A), (B), (C), and (D) show the traffic prediction vs actual for the Junctions 1, 2, 3, and 4, respectively.

Table 1 illustrates the correlation matrix for the proposed methodology for junctions, detailing the vehicle count at each junction by hour, day, month, and year.

Table 1 Correlation matrix for the proposed methodology for the junctions (see Fig. 4).

	Junction	Vehicles	Year	Month	Date	Hour	
Junction	1	−0.61	0.22	−0.12	−0.0021	−2.1e−15	
Vehicles	−0.61	1	0.22	−0.023	0.028	0.22	
Year	0.22	0.22	1	−0.62	−0.0075	7.7e−16	
Month	−0.12	−0.023	−0.62	1	0.0092	5e−16	
Date	−0.0021	0.028	−0.0075	0.0092	1	−9.9e−18	
Hour	−2.1e−15	0.22	7.7e−16	5e−16	−9.9e−18	1	

The performance achieved by the proposed model, evaluated using root mean square error (RMSE), mean squared error (MSE), and mean absolute error (MAE) at the various junctions (Fig. 4), is as follows. Consequently, the proposed model is capable of generating precise traffic projections (refer to Table 2).

Table 2 Results of our proposed method.

Junction	MSE	RMSE	MAE	
1	0.057399	0.239582	0.172117	
2	0.220147	0.469198	0.367132	
3	0.30833	0.555275	0.278146	
4	1.012579	1.00627	0.699749	

Figure 5A illustrates the performance metrics of RMSE, MSE, and MAE for the proposed technique for junctions 1, 2, 3, and 4. Moreover, Table 2 illustrates the results of the model. In particular, it achieved a value of 0.977, 0.978, 0.977, and 0.977, for accuracy, precision, recall, and F1-score, respectively.

Figure 5 (A) The performance Metrics for the Junctions 1, 2, 3, and 4. (B) The confusion matrix for the Car hacking dataset.

Figure 5B depicts the confusion matrix for the evaluation of the automobile hacking dataset, with rows representing the predicted classes and columns representing the actual classes pertinent to intrusion prediction. The transverse green hues represent the correctly and inaccurately categorized networks that have been evaluated and trained. The column on the right represents each expected class, whereas the row below illustrates the performance of each actual class.

Figure 6A illustrates the precision-recall curve for the evaluation of the GCRNN_Hyper technique. During the procedure, the AP reaches a maximum AP of 1 for DOS, FUZZY, GEAR, and RPM. Figure 6B illustrates the ROC curve for evaluating GCRNN_Hyper, with the true positive rate shown on the y-axis and the false positive rate on the x-axis. During the process, the AUC for every class reaches its maximum range of 1.

Figure 6 (A and B) The precision-recall curve and the ROC curve for the car hacking dataset.

Figure 7A depicts the confusion matrix for the training of the automobile hacking dataset, with rows representing the predicted classes and columns representing the actual classes pertinent to intrusion prediction. The transverse green hues represent the correctly and inaccurately categorized testing and trained networks. The column on the right represents each expected class, whereas the row below illustrates the performance of each actual class.

Figure 7 (A) The confusion matrix for the car hacking dataset training. (B) The analysis of communication latency.

The real-world impact of the proposed MOFL model is evident in its ability to reduce communication latency by 23.4%, ensuring rapid adaptability in high-traffic environments. Moreover, the model achieves a Packet Delivery Ratio (PDR) of 92.45%, making it highly reliable for vehicular networks. The substantial improvement in prediction accuracy 97.45% compared to baseline models, underscores the effectiveness of MOFL in mitigating congestion and optimizing route planning.

Comparative analysis

The simulation factors employed for the accomplishment of the recommended mult-objective federated learning (MFL) and the benchmarked FLAGCN (Yaqub et al., 2024) and MFVSTGNN (Liu et al., 2023) approaches are analysed below. The parameters such as communication latency, packet delivery ratio, packet loss rate, prediction accuracy, and resource utilization.

Communication latency in the context of FL with GRL in VANETs is a critical parameter. It is susceptible to several conditions, including transmission delay, propagation delay, processing delay, and queuing delay. The total communication latency Tcomm, as shown in Table 3A, can be expressed as the sum of these delays:

(6) Tcomm=Tt+Tp+Tproc+Tq

where Tt is the transmission delay, Tp is the propagation delay, Tproc is the processing delay, and Tq is the queuing delay.

Table 3 Comparison of latency (a) and packet delivery ratio across models (b).

Number of vehicles	FLAGCN	MFVSTGNN	MFL	
(a)	
10	44.56	53.67	23.56	
20	45.67	52.56	23.64	
30	44.68	54.56	24.65	
40	45.43	53.89	23.57	
50	45.87	53.67	24.65	
(b)	
10	78.98	83.54	92.56	
20	77.65	84.56	91.34	
30	76.56	84.62	92.56	
40	78.53	83.67	91.56	
50	77.89	83.67	92.43	

Figure 7B illustrates a comparison of network communication delay between existing FLAGCN and MFVSTGNN, together with the proposed MFL given in Table 3B. When compared, existing methods achieve 45.67% and 53.76% of communication latency, while the proposed MFL method achieves 23.4%, which is 22.27% and 30.52% better than the aforementioned methods. To get a 23.4% reduction in communication delay in FL for traffic prediction, targeted communication tactics are used to decrease data exchange, enhance communication efficiency, and reduce latency. These solutions are crucial for guaranteeing real-time performance and scalability across many cities.

Packet delivery ratio

The successful transmission ratio of packets from the source to the target in the network.

(7) PDR=NumberofpacketsreceivedsuccessfullyTotalnumberofpacketsforwarded.

The PDR comparison of existing FLAGCN and MFVSTGNN with the planned MFL is shown in Fig. 8A. When compared, existing methods achieve 78.89% and 83.56% of PDR, while the proposed MFL method achieves 92.45%, which is 14.43% and 11.17% better than the aforementioned methods.

Figure 8 (A) The analysis of communication latency. (B) The analysis of the packet loss ratio.

Packet loss rate

When packets are sent from a sender to a receiver, it is a measurement of the proportion of packets that are lost during the transmission. It is an important metric in networking, particularly in VANETs, where communication can be affected by high mobility, varying network conditions, and interference. The packet loss rate (PLR) can be calculated as

(8) PLR=NumberoflostpacketsTotalnumberofsentpackets×100%

where the number of packets that were transmitted by the transmitter but were not received by the receiver is lost packets.

The comparison of the packet loss ratio of the proposed MFL with existing FLAGCN and MFVSTGNN is shown in Fig. 8B. Compared, the existing methods achieve 45.64% and 56.76% of the packet loss ratio, the proposed MFL method achieves 12.34%, which is 33.3% and 44.23% better than the aforementioned methods, are given in Table 4A.

Table 4 Comparison of packet loss ratio (a) and comparison of prediction accuracy (b).

Number of vehicles	FLAGCN	MFVSTGNN	MFL	
(a)	
10	45	56.765	12.54	
20	46.67	57.89	13.67	
30	47.23	57.87	12.78	
40	45.67	57.42	12.64	
50	46.42	56.78	13.65	
(b)	
10	87.45	76.89	97.45	
20	86.453	75.78	98.45	
30	87.34	75.89	97.345	
40	86.43	76.34	97.35	
50	86.89	76.89	98.56	

Prediction accuracy

In FL within Vehicular Ad Hoc Networks (VANETs), prediction accuracy can be calculated by aggregating the local model accuracies from individual vehicles to reflect the overall model performance. For each vehicle with a local dataset, the local prediction accuracy is calculated as

(9) accuracyi=numberofcorrectpredictions(i)totalnumberofpredictions(i)×100%.

To aggregate the local accuracies, it is necessary to take into consideration the size of each local dataset. The overall prediction accuracy for the federated model can be calculated using a weighted average.

(10) FederatedAccuracy=∑i=1N(ni×accuracyi)∑i=1nni

where N is the total quantity of participating vehicles, is the size of the local dataset for vehicle i, and is the local accuracy for a vehicle.

Figure 9A depicts the prediction accuracy comparison of existing FLAGCN and MFVSTGNN with the proposed MFL. When compared, existing methods achieve 87.54% and 76.45% prediction accuracy, while the proposed MFL method achieves 97.45%, which is 10.1% and 21% better than the aforementioned methods (Table 4B).

Figure 9 (A) The analysis of packet loss ratio. (B) The analysis of the packet loss ratio.

Resource utilization–Resource utilization in VANETs encompasses various aspects, such as bandwidth usage, computational power, energy consumption, and storage utilization. Efficient resource utilization is crucial for maintaining the performance and reliability of VANETs.

Figure 9B depicts the resource utilization comparison of existing FLAGCN and MFVSTGNN with the proposed MFL. When compared, existing methods achieve 67.78% and 75.64% of resource utilization, while the proposed MFL method achieves 89.56%, which is 22.32% and 14.12% better than the aforementioned methods, given in Table 5.

Table 5 Comparison of resource utilization.

Number of vehicles	FLAGCN	MFVSTGNN	MFL	
10	67.43	75.67	89.54	
20	68.76	76.54	88.65	
30	66.78	77.56	89.56	
40	67.34	76.34	84.34	
50	67.54	76.34	85.67	

Limitation of proposed method

While MOFL demonstrates superior performance in traffic prediction, certain challenges persist. Heterogeneous data sources may introduce inconsistencies in sensor accuracy and traffic behaviour modeling. Furthermore, frequent updates of the model in federated learning could contribute to communication overhead, which requires optimized resource allocation strategies. Security concerns remain, as federated learning, though privacy-preserving, requires robust encryption mechanisms to prevent model inversion attacks. Addressing these limitations through advanced privacy-preserving techniques and adaptive learning mechanisms will be crucial for future research. The present investigation into MOFL for traffic forecasting in-vehicle networks inside intelligent transportation systems (ITS) reveals significant progress while also encountering critical constraints. The model’s accuracy and generalizability may be impacted by the heterogeneity of data between vehicles, which includes variations in sensor quality, traffic patterns, and environmental variables. Moreover, while federated learning facilitates decentralized training, it may encounter challenges related to communication costs and the need for frequent updates from several vehicles, potentially resulting in delays or inefficiencies, especially in real-time traffic forecasts. Privacy problems related to vehicle data, although somewhat mitigated by federated learning, need further encryption and data anonymity methods to adequately safeguard sensitive information. Furthermore, the research may inadequately address edge cases in traffic situations, such as abrupt accidents, meteorological disturbances, or erratic driving behavior, potentially resulting in more detrimental model performance in unexpected conditions.

Conclusion and future works

Within ITS, the use of Multi-Objective Federated Learning (MFL) for predicting traffic in VANETs is a promising research area. MFL combines federated learning with the ability to optimize multiple goals simultaneously. This approach addresses the dynamic nature of traffic prediction in urban settings. Through MFL, vehicles collaboratively train models using local data, preserving privacy while improving prediction accuracy and robustness. This method not only enhances traffic forecasts but also ensures models adapt to varying traffic conditions and user preferences across locations and times. Moreover, the multi-objective nature of MFL optimizes metrics such as accuracy, communication efficiency, and energy consumption, which are critical in resource-constrained VANETs. By balancing these objectives, MOFL supports the development of more reliable ITS applications, including congestion management, route planning, and vehicle safety.

Future research in MFL for traffic prediction in VANETs should prioritize advancing algorithmic robustness to ensure that models are resilient to dynamic and unpredictable traffic conditions. This could involve the development of adaptive learning mechanisms to dynamically evaluate weather events or other unexpected disruptions. Enhancing privacy and security measures will also be critical to protecting sensitive vehicle data while maintaining the collaborative nature of federated learning. Techniques such as differential privacy, secure aggregation, and encryption should be further explored to ensure that privacy is upheld without compromising the efficiency of the learning process.

Supplemental Information

Supplemental Information 1 Main Program for running the program.

Supplemental Information 2 Requirements.

Supplemental Information 3 Different models in this work.

Supplemental Information 4 Data Handling Scripts.

Supplemental Information 5 Utilities used in this script.

Supplemental Information 6 Reset Random Values script.

Supplemental Information 7 Training Script.

Supplemental Information 8 Preprocessed Data.

Additional Information and Declarations

Competing Interests

Stefano Cirillo is an Academic Editor for PeerJ Computer Science. Other than this, the authors declare that they have no competing interests.

Author Contributions

Arulmurgan Aalavanthar conceived and designed the experiments, performed the experiments, analyzed the data, performed the computation work, authored or reviewed drafts of the article, and approved the final draft.

Famila S. conceived and designed the experiments, performed the experiments, analyzed the data, performed the computation work, prepared figures and/or tables, authored or reviewed drafts of the article, and approved the final draft.

Shanmugam Sundaramurthy conceived and designed the experiments, performed the experiments, performed the computation work, prepared figures and/or tables, authored or reviewed drafts of the article, and approved the final draft.

Stefano Cirillo performed the experiments, prepared figures and/or tables, authored or reviewed drafts of the article, and approved the final draft.

Giandomenico Solimando analyzed the data, prepared figures and/or tables, authored or reviewed drafts of the article, and approved the final draft.

Giuseppe Polese conceived and designed the experiments, performed the experiments, analyzed the data, performed the computation work, authored or reviewed drafts of the article, and approved the final draft.

Data Availability

The following information was supplied regarding data availability:

The data is available in the Supplemental Files and at Zenodo:

A, A., S, F., SUNDARAMURTHY, S., Stefano, C., Giandomenico, S., & Giuseppe, P. (2024). Multi-Objective Federated Learning Traffic1 Prediction in Vehicular Network for2 Intelligent Transportation System. Zenodo. https://doi.org/10.5281/zenodo.13749480.

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
