# Peer review of "Multi-objective federated learning traffic prediction in vehicular network for intelligent transportation system"

_PeerJ Computer Science, doi:10.7717/peerj-cs.2922_

## Round 0.1 · original submission · Major Revisions

Based on the external reivews, a major revision is needed. The authors should clearly addtess all the metioned issues for further consideration.

Reviewer 1 ·

Basic reporting

The increasing growth of urban freight traffic poses challenges for traffic management systems. To address these challenges, the authors of the paper propose an approach to traffic prediction through multi-objective federated learning. This method uses decentralized data processing to improve the accuracy of traffic speed forecasts.

-The paper has clear and unambiguous professional english throughout. Technical terms and concepts are explained well and accessible to readers. The authors maintain a consistent tone and language style.
-The authors provide a comprehensive review of existing literature but it lacks literature related to traffic prediction, federated learning, and reinforcement learning. There must be different subsections for each clearly.
-The paper is well-structured
-

Experimental design

The experiment results are well described.

Validity of the findings

FIndings are valid.

Additional comments

-What are the main advantages of using Multi-Objective Federated Learning for traffic prediction? It is not clear and lacks motivation for the work.
-How does reinforcement learning enhance dynamic decision-making in traffic prediction scenarios?
- explain what is the significance of integrating graphical models in traffic intensity identification.
-How are the performance metrics used to evaluate the proposed traffic prediction method?
- experiment results show a high accuracy of the proposed work? therefore, there is a need to present a confusion metrics and false positive ratios.
-How is data privacy maintained when using Federated Learning for traffic data from multiple cities? need justification and clarification within the paper regarding the work.
-What and why communication strategies are implemented to achieve a 23.4% communication delay reduction?
-How does the prediction accuracy of 97.45% compare with existing traffic prediction methods? what is the F-1 score and Recall value?
-What limitations exist in the current study, and how can future research address them?

Reviewer 2 ·

Basic reporting

This paper introduces a new approach for traffic prediction using multi-objective federated learning. Several participants implement collaborative training instead of relying on a centralized cloud server for data processing.
The article is well-structured and organized. 
The contribution to the related field is sufficient. 
It is recommended that more details be added to the presented comparative analysis to highlight the advantages of the proposed model.
The authors should describe the research results from a multidimensional perspective.
Please also highlight the advantages of the proposed approach in the introduction. 
Please also add details about the communication load.
How the hyperparameters are selected and why they should be highlighted.
How did the authors select the convergence criteria and iteration rounds?
Why does adding vehicles not have a significant adverse effect? Please also consider a higher number of vehicles.
Please also add convergence curves.
It would be better to show the result; the current format with bar charts is not clear.

Experimental design

no comment

Validity of the findings

no comment

Additional comments

There are no additional comments.

---

## Round 0.2 · Minor Revisions

Based on the external review, a minor revision is needed before final acceptance.

Reviewer 2 ·

Basic reporting

Thank you for the modifications.

The resolution of the newly added images (Figures 3 and 4) is lower than expected. Please provide higher-quality versions.

Additionally, I expected the authors to improve the explanation of the main problem and clarify the application of the proposed methods. This version remains too brief, and more detail is needed in these two areas.

Experimental design

Good

Validity of the findings

Good

---

## Round 0.3 · Minor Revisions

Based on the external reviews, a minor revision decision has been made. The authors should revise the manuscript accordingly for further considerations.

Reviewer 2 ·

Basic reporting

Thank you for the modification. There are no further comments.

Experimental design

Valuable and well presented

Validity of the findings

Yes

Additional comments

No

Reviewer 3 ·

Basic reporting

The paper presents a ML-based approach for traffic prediction, using several advanced techniques such as multi-objective federated learning.

Following the introduction, authors present related work; they then describe the basics of the proposed approach, and the proposed approach itself. Then, authors describe the respective evaluation. Authors also discuss the limitations of the proposed approach and, finally, draw their conclusions.

Major comments follow.

- Overall, the paper is well written, presents an important topic with remarkable results. My comments are only minor. The most important one is to strengthen the link between the overall approach and its different components. A short, high-level overview (plus some connecting sentences here and there) could be sufficient.

Detailed comments follow.

- I appreciated the clarity of related work. The authors briefly describe the issue faced by each paper (e.g., freight traffic speed prediction for the second paper). This clever description is only missing in the third surveyed paper (Shen at al., 2024): I would suggest to the same for that paper as well.
- I would suggest to improve the pattern matching description at line 186. In detail, I would only suggest to better “introduce” this sentence, since it is (at least initially) totally unrelated to weights.
- I would suggest to improve section 4.2 by better connecting it with the overall approach. In other words, the description in itself is clear, but it is not entirely clear how it relates to the overall approach.

Minor comments follow.

- I would suggest to explain the meaning of “RSU”. This suggestion applies also to other non-obvious acronyms.
- I would suggest to harmonize the colors of histogram plots.

Experimental design

The experiments evaluate the proposed approach along several axis, including comparisons with literature. The only suggestion I have to is to harmonize the plots (same color, font, etc).

Validity of the findings

Findings are adequately discussed.

Additional comments

N.A.

---

## Round 0.4 · accepted · Accept

Based on the external review, it can be accepted.

Reviewer 3 ·

Basic reporting

no comment

Experimental design

no comment

Validity of the findings

no comment

Additional comments

Thanks for addressing my comments. I do not have additional concerns.